# Complete continuum of maternity care and associated factors among mothers who gave birth in the last twelve months in Mekane Selam town North-East Ethiopia: A community-based cross-sectional study,2021

Hibist Tilahun Mengistie[1], Mengistu Abate Belay[2], Abrham Debeb Sendekie[3]*, Anguach Shitie[2], Dagne Addisu Sewyew[4]

1 Amhara Regional State Health Bureau Merawi Primary Hospital, Merawi, Ethiopia, 2 College of Medicine and Health Sciences School of Nursing, Wollo University, Dessie, Ethiopia, 3 College of Medicine and Health Sciences, Wachemo University, Hossana, Ethiopia, 4 College of Health Sciences, Debre Tabor University, Debre Tabor, Ethiopia

* abrishdebeb86@gmail.com

## Abstract

### Background

Complete continuum of maternity care has significant value in improving the health and well-being of mothers and newborns. Assessing the complete continuum of maternity care is a global priority, particularly in developing countries. Despite the fact that the complete continuum of maternity care prevents more than half of all maternal and neonatal deaths, Ethiopia remains one of the largest contributors to the global burden of maternal and neonatal deaths due to the low implementation of the complete continuum of maternity care. Thus, this study aimed to assess the coverage of the complete continuum of maternity care and associated factors among mothers who gave birth in the last 12 months in the study area.

### Methods

A community-based cross-sectional study with a multistage sampling technique was conducted among 479 mothers who gave birth in the last 12 months in Mekane Selam town. Data were collected from September 1 to November 30/2021 using an interviewer-administered questionnaire. Binary logistic regression analysis was computed. In multivariable logistic regression analysis, an adjusted odds ratio (AOR) with a 95% confidence interval (CI) and a P- value< 0.05 were used to identify significantly associated factors.

### Results

The coverage of a complete continuum of maternity care was 42.4% (95% CI: 37.9%, 47%). Respondents with higher educational status (AOR = 4.17, 95% CI: 1.52, 11.44), pre-

**Data Availability Statement:** All relevant data are within the paper and its Supporting Information files.

**Funding:** The author(s) received no specific funding for this work.

**Competing interests:** The authors have declared that no competing interests exist.

pregnancy contraception utilization (AOR = 3.53, 95% CI: 1.80, 6.92), planned pregnancy (AOR = 2.97, 95% CI: 1.27, 6.97) and early initiation of antenatal care (AOR = 4.57, 95% CI: 2.86, 7.31) were significantly associated with complete continuum of maternity care.

## Conclusion

The coverage of complete continuum of maternal care was low in the study area. The coverage could be expanded by making the necessary interventions on the associated factors. It is essential for women to acquire education, utilize contraception, plan their pregnancies, and begin antenatal care at an early age in order to enhance the coverage of complete continuum of maternity care.

## Introduction

Continuum of care in its time dimension refers to a situation where a woman and her children receive maternal, newborn, and child health (MNCH) services from pre-pregnancy, pregnancy, childbirth, and postpartum to childhood. The place dimension focuses on integrated service delivery provided at the household level, community level, and health facility level [1]. Continuum of care is one of the strategies to monitor progress in the improvement of MNCH services and can be avoided more than half of maternal and neonatal deaths by applying an effective and low-cost complete continuum of maternity care [2].

According to a 2019 report by the World Health Organization (WHO) and the United Nations International Children's Emergency Fund (UNICEF), approximately 295,000 women die worldwide each year from pregnancy or childbirth-related complications, with 94% of these deaths occurring in low- and lower-middle-income countries and nearly two-thirds (65%) occurring in Sub-Saharan Africa [3]. Ethiopia, India, Pakistan, Nigeria, and the Democratic Republic of the Congo are the five nations that account for half of all newborn fatalities worldwide, according to UNICEF's 2017 report [4].

According to the 2016 Ethiopia Demographic and Health Survey (EDHS), Ethiopia has a high rate of maternal mortality, which is the health sector's single most serious concern. Maternal mortality was expected to be 412 deaths per 100,000 live births, with neonatal mortality at 29 deaths per 1000. Nearly three-quarters of all maternal deaths (75%) are caused by direct obstetric causes that can be avoided with cost-effective measures [5]. The continuum of care is advocated as a core framework for the delivery of MNCH services, linking critical interventions across the pregnancy, birth, and postpartum stages in order to meet the Sustainable Development Goal (SDG) [6]. If proper health interventions are provided during birth and the first week of life, it is predicted that around 80% of maternal deaths and up to two-thirds of newborn deaths might be prevented globally [7]. One of the suggestions in the SDG to accomplish the global aim of the reduction of the global maternal mortality ratio to at least <70 per 100 000 live births or no country has no more than 140 maternal mortality ratio, reducing neonatal deaths to 12 per 1000 live births, and under-five deaths to <25 per 1000 live births by the year 2030 was to improve completion along the continuum of care for MNCH services [8].

According to studies from around the world, women's completion of the continuum of care for maternity and child health services is very low, and women do not access MNCH services serially along the spectrum of care. According to the data, the magnitude of the continuum of maternity care was 25% in South Asia, 14% in Sub-Saharan Africa, 60% in Cambodia, and 8% in Ghana among women who were receiving a full range of important components of

maternal health services [9]. South Asia and Sub-Saharan Africa are two of the world's regions in desperate need of reforms in the continuum of care [10]. According to a study based on EDHS 2016 data, only 9.1% of Ethiopian women completed the continuum of maternity care [11]. The magnitude of the complete continuum of maternity care ranges from 9.7% in Arba Minch Zuria Woreda [12], 12.1% in West Gojjam Zone [13], 21.6% in Gondar Zuria and Dabat districts [14], and 37.2% in Debre Berhan town, according to various studies conducted in Ethiopian rural districts [15]. The prevalence of ANC in the component of complete continuum of care reached 43%, skill birth delivery reached 50%, and PNC was 34%, according to the Ethiopian Mini Demographic and Health Survey (EMDHS). [16]. A lack of competent treatment at any point along the full continuum of care is connected to poor maternal health outcomes [17]. Lack of formal education and healthcare access issues such as distance to health facilities, lack of media exposure, unexpected pregnancy, and late commencement of ANC were found to be variables affecting the full continuum of maternity care [10, 18, 19].

Even though there have been some studies on the complete continuum of maternity care in Ethiopia, they have not been adequately investigated there, and no prior studies on the topic have been done in our study setting. There hasn't been quite enough intervention in the continuum of maternity care as a result of these deficiencies. Determining the coverage of a complete continuum of maternity care and associated factors among women who gave birth in the previous 12 months in Mekane Selam, Ethiopia, was the aim of this study as a result.

## Methods

### Study design and setting

A community-based cross-sectional study was undertaken in Mekane selam Town, North-East Ethiopia, from September 1 to November 30, 2021. The town of Mekane Selam is located in the Amhara Regional State of North-East Ethiopia. It is 467 kilometers from Ethiopia's capital, Addis Ababa, and 346 kilometers from Bahir Dar, the capital of the Amhara regional state. According to the statistics gathered from town administrative authorities at the time this study was conducted, the town's total population is anticipated to be 120,798 people, with 62,965 females and 42,452 women between the ages of 15 and 49. Mekane Selam town currently manages seven kebeles. There is also a public hospital, a public health center, five health posts, and numerous medium and higher private clinics in Mekane Selam.

### Participants

The source populations were all women who gave birth in the last 12 months in Mekane Selam town, whereas the study populations comprised all women who gave birth in the last 12 months at selected Kebeles in Mekane Selam town North-East Ethiopia during the data collection period. All women who gave birth in the last 12 months at selected households in Mekane Selam town, North East Ethiopia, during the data period were included. Based on the zonal department health office and health extension workers' information reports an overall 1250 women who gave birth in the last 12 months in Mekane Selam town, North East Ethiopia, and 722 women who gave birth in the last 12 months at selected Kebeles in Mekane Selam town North-East Ethiopia.

### Sample size determination and sampling procedure

First, the sample size was determined by using a single population proportion formula by considering P = 12.1% women's retention on the continuum of the maternal care pathway in West Gojjam Zone [13], Zα/2 at 95% confidence level, 5% margin error (d), 10% non-response rate

and by assuming 1.5 design effect we get the minimum sample size for the first objective was 269 but, The required minimum sample size for this study was 479, as determined by the second objective using Epi info software version 7.2 with consideration of factors affecting complete continuum of maternity care by considering 95% of confidence interval, 80% power, odds ratio 3.96, 10% none response rate, and 1.5 design effects.

The number of women who gave birth in each Kebele was derived from the report of the zonal health office and health extension workers. The requisite samples were chosen and proportionally allocated to each selected kebele (Fig 1).

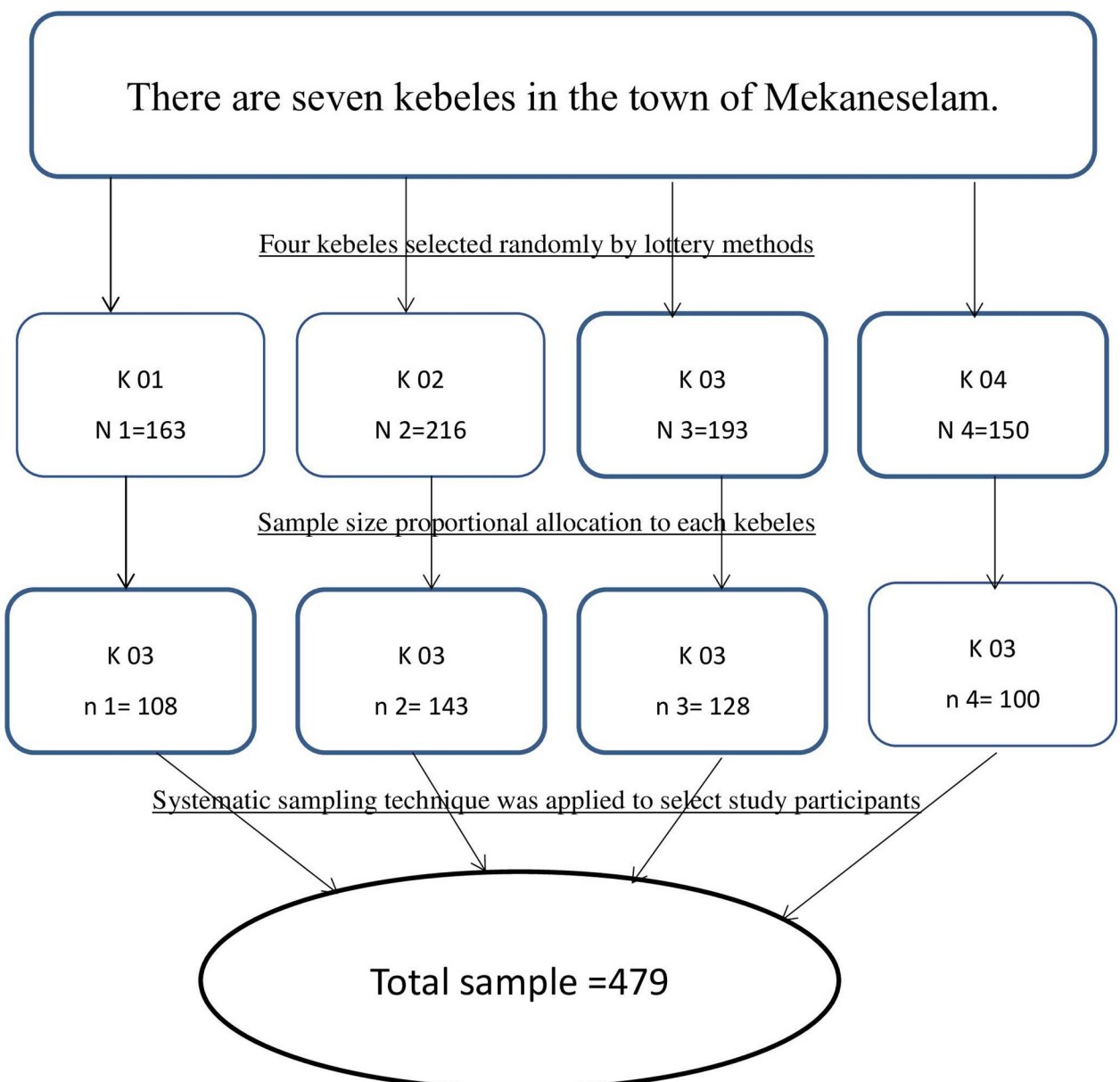

**Fig 1. Schematic representation of sampling procedure in Mekane Selam town kebeles.** Where N = total study population in each kebele and n = total sample size in each kebele.

## Variables of the study

**Dependent variable:** Complete continuum of maternity care (Yes/No)

## Independent variables:

**Socio-demographic variables:** Age, marital status, religion, ethnicity, educational status of the mother, educational status of the husband, occupation of the mother, and occupation of the husband.

**Maternal health care services related variables:** Distance from a health facility, knowledge of danger signs of pregnancy, exposure to mass media, and women's autonomy in healthcare decision making.

**Obstetric related variables:** Parity, pre-pregnancy contraception utilization, planned pregnancy, timing of first antenatal care visit, and place of delivery.

## Operational definition

**Completed continuum of maternity care:** A woman had completed a continuum of maternity care if she had four or more ANC visits in her recent pregnancy by skilled health personnel, a recent delivery assisted by a skilled birth attendant and having PNC within 48 hours after recent birth [13, 14, 19, 20].

**Women's autonomy in healthcare decision making:** In this study, autonomy in healthcare decision making was assessed by asking (Yes/no question). A woman was said to have decision making power on seeking healthcare services if she alone or with her husband decide on seeking maternal healthcare services [2, 21].

**Timing for first ANC visit:** The time or gestational period at which the pregnant women first attend ANC clinic was classified as within 16 weeks or after 16 weeks [13, 15].

**Knowledge on key pregnancy danger signs:** Women were classified as knowledgeable if they mentioned at least two of the key danger signs of pregnancy (vaginal bleeding, severe headache, blurring of vision, and swelling of legs or face), or if not, they were classified as not knowledgeable [7, 12].

**Planned pregnancy:** Planning for pregnancy was measured by asking (Yes/no question) a woman whether the recent pregnancy occurred when she had desired for child birth or not.

**Exposure to mass media:** Exposure to mass media was assessed by asking (Yes/no question) to women's habit of reading a newspaper, watching TV, or listening to radio to access relevant information on maternal and child health services [7].

## Data collection tools and quality managements

A structured face-to-face interviewer administered questionnaire developed from a prior study was used to collect data [2, 7, 10, 12, 13, 19, 22–28]. English version was translated to Amharic version by two native speakers of Amharic language (one was reproductive health and the other was English language and literatures in professions). Then back translation to English was done by another two individuals who could speak English (again one was from public health and the other from English language and literatures). Individuals involved in translations were those who knew local says for some expressions. Final questionnaire was prepared by involving both groups (translators) after resolving inconsistences via discussion for some meanings and terminologies. The tool was piloted on 5% (24) pregnant women in Akesta town where actual study population are culturally related. Amendment was made by the investigators. Data was collected by four trained midwives and supervised by two BSc midwifery professionals. The training was given for two days on the concepts of the questionnaire related

to the objectives. Role-play was made during training on how to approach study participants ethically and make interviews consistently without disrupting the concepts. Comments were given by data collectors, supervisors and principal investigator immediately upon completion of the role-play. On a daily basis, supervisors and principle investigators double-checked and summarized the acquired data for completeness and consistency.

## Data processing, analysis, and interpretation

First, data were checked for completeness, and consistency, coded manually, and then entered to Epi Data version 4.2 and exported to SPSS (Statistical Package for Social Science) version 25 for cleaning and analysis. Descriptive statistics were expressed in frequency and percentage. The data was presented using text, tables, and graphs. A binary logistic regression model was fitted to identify statistically significant independent variables. Initially, bivariable logistic regression analysis was performed between the dependent variables and each of the independent variables in sequence. All independent variables from a bivariable analysis with a p-value less than 0.2 were entered into multivariable analysis for further analysis and to adjust for confounding variables and a backward likelihood ratio was used. Model fitness was checked by the Hosmer-Lemeshow goodness of test (P-value = 0.88) and multicollinearity between the explanatory variables was checked using the variance inflation factor (VIF>10). In multivariable logistic regression analysis, statistically significant was declared at P-value <0.05, and the strength of association between the outcome variables and independent variables was reported by using the AOR with their 95% CI.

## Ethical considerations

The study was conducted after the confirmation of national and international ethical guidelines for biomedical research involving human subjects. Before collection of data, ethical clearance obtained from institutional review board (IRB) of Wollo University. Then permission letter was received from regional, zonal and district health office. Study participants were informed about how they were included in the study, the purpose of the study, their rights to withdraw or continue and potential benefits and harms of the study. Study participants were also told that the information they provide will be used only for the research purpose and will not be disclosed for anyone including during publication. Written consent form was prepared and attached together with each questionnaire to obtain approval from each study participant by signature.

## Results

### Socio-demographic characteristics of the study participants

The interviewer-administered questionnaire was completed by 469 study participants, yielding a response rate of 97.9%. The women average age was 26.7 (SD 4.89) years. More than half of the participants, 242 (51.6%), were between the ages of 25 and 34. The Orthodox religion was represented by 264 (56.3%) of the respondents. In addition, 140 (29.9%) of women had completed primary school, and 186 (39.7%) of women were housewives (Table 1).

### Maternal health care service related characteristics of the study participants

309 (65.9%) of women had access to mass media. Among the study participants, 365 (77.8%) of women reported that the average time to reach health facilities by foot was less than thirty

**Table 1. Socio-demographic characteristics of study participants in Mekane Selam town NorthEast Ethiopia 2021, (n = 469).**

| Variable | Category | Frequency | Percentage (%) |
|---|---|---|---|
| **Age of respondents in years** | 15–24 | 186 | 39.7 |
| | 25–34 | 242 | 51.6 |
| | 35–49 | 41 | 8.7 |
| **Religion of respondents** | Orthodox | 264 | 56.3 |
| | Muslim | 179 | 38.2 |
| | Protestant | 23 | 4.9 |
| | Catholic | 3 | 0.6 |
| **Marital status** | Married | 462 | 98.5 |
| | Single/ divorced/ widowed | 7 | 1.5 |
| **Ethnicity of respondents** | Amhara | 459 | 97.9 |
| | Others* | 10 | 2.1 |
| **Educational status of respondents** | No education | 91 | 19.4 |
| | Primary | 140 | 29.9 |
| | Secondary | 112 | 23.8 |
| | Higher | 126 | 26.9 |
| **Educational status of husband** | No education | 46 | 10 |
| | Primary | 111 | 24 |
| | Secondary | 148 | 32 |
| | Higher | 157 | 34 |
| **Occupation of respondents** | House wife | 186 | 39.7 |
| | Employed | 151 | 32.2 |
| | Merchant | 120 | 25.6 |
| | Daily laborer | 12 | 2.5 |
| **Husband occupation** | Employed | 185 | 40 |
| | Merchant | 223 | 48.3 |
| | Daily laborer | 39 | 8.5 |
| | Others** | 15 | 3.2 |

Others*: Oromo and Tigre, others

**: farmer and driver

minutes. 383 (81.7%) of the respondents had autonomy in health care decision making and 187 (39.9%) of the women were knowledgeable about danger signs of pregnancy (Table 2).

**Table 2. Maternal health care services related characteristics of the study participants in Mekane Selam town North-East Ethiopia, 2021 (n = 469).**

| Variable | Category | Frequency | Percentage (%) |
|---|---|---|---|
| **Exposure to mass media** | Yes | 309 | 65.9 |
| | No | 160 | 34.1 |
| **Perceived required time to reach health facilities by foot** | <30 minutes | 365 | 77.8 |
| | ≥30 minute | 104 | 22.2 |
| **Autonomy to health care decision making** | Yes | 383 | 81.7 |
| | No | 86 | 18.3 |
| **Knowledge of danger signs of pregnancy** | Knowledgeable | 187 | 39.9 |
| | Not knowledgeable | 282 | 60.1 |

## Obstetrics related characteristics of study participants

Sixty-four (13.6%) of the study participants were grand multipara mothers, 374 (79.7%) of the study participants used pre-pregnancy contraceptive methods, and 79.3% of women reported that their recent pregnancy was planned. Four hundred twenty-nine (91.5%) of the participants had antenatal care visits, more than half of the 220 (51.3%) had fourth and above antenatal care visits in their recent pregnancy, and 186 (43.4%) of the study participants were receiving their first antenatal care within 16 weeks of gestation. About four hundred nineteen (89.3%) of the study participants were delivered at health facilities. Three hundred thirty-seven (71.9%) of the study participants had postnatal care visits and 281 (83.4%) of them were receiving their postnatal care visit within 48 hours of birth (Table 3).

## Coverage of the complete continuum of maternity care

In this study, the coverage of a complete continuum of maternity care among women who gave recent births in the last 12 months preceding the study was 42.4% (95% CI: 37.9%, 47%). Among the participants, 51.3% of the women had received four and above ANC visits in their recent pregnancy and only 46.9% of them were continued on the pathway and attended by a skilled health provider at delivery and 42.4% of them had received PNC within 48 hours after birth (Fig 2).

**Table 3. Obstetrics related characteristics of study participants in Mekane Selam town North-East Ethiopia, 2021 (n = 469).**

| Variable | Category | Frequency | Percentage (%) |
|---|---|---|---|
| Parity | 1–2 | 264 | 56.3 |
|  | 3–4 | 141 | 30.1 |
|  | ≥5 | 64 | 13.6 |
| Pre-pregnancy utilization of contraceptive methods | Yes | 374 | 79.7 |
|  | No | 95 | 20.3 |
| Was the pregnancy planned | Yes | 372 | 79.3 |
|  | No | 97 | 20.7 |
| Have ANC visit | Yes | 429 | 91.5 |
|  | No | 40 | 8.5 |
| Timing of 1st ANC visit | Within 16 weeks | 186 | 43.4 |
|  | After 16 weeks | 243 | 56.6 |
| Number of ANC visits | 1 | 32 | 7.4 |
|  | 2–3 | 177 | 41.3 |
|  | ≥4 | 220 | 51.3 |
| Place of delivery | Health facility | 419 | 89.3 |
|  | Home | 50 | 10.7 |
| Birth attendant during delivery | Health professional | 419 | 89.3 |
|  | Traditional birth attendant | 44 | 9.4 |
|  | Others | 6 | 1.3 |
| Have PNC visit | Yes | 337 | 71.9 |
|  | No | 132 | 28.1 |
| Timing of PNC visit | Within 48 hours | 281 | 83.4 |
|  | 49–72 hours | 24 | 7.1 |
|  | 73 hours-7 days | 15 | 4.5 |
|  | 8–42 days | 17 | 5 |

Others: relative/ friend

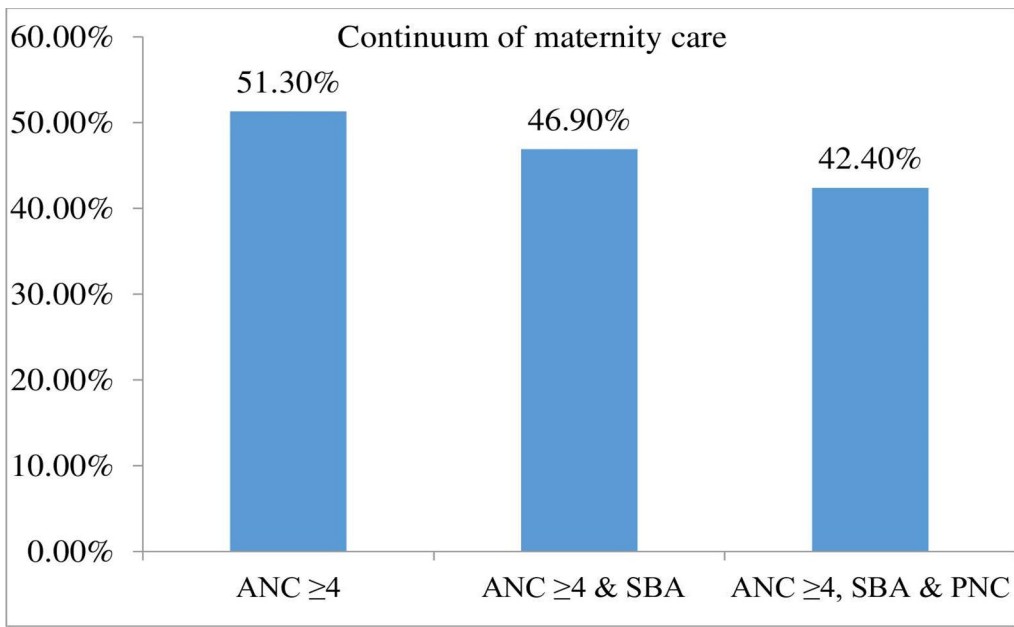

**Fig 2. Continuum of maternal health care services among women who gave birth in the last 12 months in Mekane Selam town North-East Ethiopia, 2021.**

### Factors associated with a complete continuum of maternity care

Bivariable and multivariable logistic regression analyses were done to identify factors associated with a complete continuum of maternity care. In the bivariable logistic regression analysis: age of respondents, educational status of respondents, educational status of the husband, occupation of respondents, occupation of husband, exposure to mass media, distance from health facilities, autonomy in health care decision making, knowledge of danger signs of pregnancy, parity, pre-pregnancy contraception utilization, planned pregnancy, and timing of first ANC visit were associated with a complete continuum of maternity care at a p-value of <0.2 and found to be candidate variables for multivariable logistic regression analysis. In the multivariable logistic regression analysis: the educational status of respondents, pre-pregnancy contraception utilization, planned pregnancy, and timing of first ANC visit were statistically significant with a complete continuum of maternity care.

The odds of a complete continuum of maternity care were 4 times higher among respondents with higher educational status (college and above) compared to those who had no formal education (AOR = 4.17, 95% CI: 1.52, 11.44). The odds of a complete continuum of maternity care were 3.7 times higher among women with secondary educational status compared to those who had no formal education (AOR = 3.69, 95%CI: 1.43, 9.50). Besides, the odds of a complete continuum of maternity care were 3 times higher among respondents with primary educational status (AOR = 3.04, 95%CI: 1.31, 7.06) compared to those who had no formal education. The odds of a complete continuum of maternity care were 3.5 times higher for those who used pre-pregnancy contraception compared to those who did not use pre-pregnancy contraception (AOR = 3.53, 95% CI: 1.80, 6.92). Women who planned for their recent pregnancy had also higher odds of completion as compared to those who did not plan for their recent pregnancy (AOR = 2.97, 95% CI: 1.27, 6.97). The odds of a complete continuum of maternity care were 4.6 times higher among mothers who initiated ANC within the first 16 weeks (AOR = 4.57, 95% CI: 2.86, 7.31) than those booked lately (Table 4).

**Table 4. Bivariable and multivariable logistic regression analysis of factors associated with complete continuum of maternity care among mothers who gave birth in the last 12 months in Mekane Selam town North-East Ethiopia, 2021(n = 469).**

| Variable | Continuum of care | | COR (95% CI) | AOR (95% CI) |
|---|---|---|---|---|
| | Yes (%) | No (%) | | |
| Age of respondents | | | | |
| 15–24 | 89 (47.8) | 97 (52.2) | 11.62 (3.46, 38.97) | 2.74 (0.45,16.34) |
| 25–34 | 107 (44.2) | 135 (55.8) | 10.04 (3.01, 33.41) | 2.41 (0.43,13.43) |
| 35–49 | 3 (7.3) | 38 (92.7) | 1 | |
| Educational status | | | | |
| No education | 12 (13.2) | 79 (86.8) | 1 | |
| Primary | 51 (36.4) | 89 (63.6) | 3.77 (1.87, 7.58) | 3.04 (1.31, 7.06) * |
| Secondary | 50 (44.6) | 62 (55.4) | 5.30 (2.60, 10.82) | 3.69 (1.43, 9.50) * |
| Higher | 86 (68.3) | 40 (31.7) | 14.15 (6.93, 28.89) | 4.17 (1.52, 11.44) ** |
| Educational status of husband | | | | |
| No education | 8 (17.4) | 38 (82.6) | 1 | |
| Primary | 20 (18) | 91 (82) | 1.04 (0.42, 2.57) | 0.61 (0.21, 1.78) |
| Secondary | 68 (45.9) | 80 (54.1) | 4.03 (1.76, 9.24) | 1.34 (0.47, 3.84) |
| Higher | 103 (65.6) | 54 (34.4) | 9.06 (3.94, 20.78) | 2.02 (0.65, 6.33) |
| Occupation of respondents | | | | |
| House wife | 56 (30.1) | 130 (69.9) | 1 | |
| Employed | 94 (62.3) | 57 (37.7) | 3.82 (2.43, 6.03) | 0.9 (0.38, 2.14) |
| Merchant | 48 (40) | 72 (60) | 1.54 (0.95, 2.50) | 1.15 (0.59, 2.24) |
| Daily laborer | 1 (8.3) | 11 (91.7) | 0.21 (0.27, 1.67) | 0.93 (0.07, 12.46) |
| Husband occupation | | | | |
| Employed | 110 (59.5) | 75 (40.5) | 9.53 (2.09, 43.47) | 1.99 (0.31, 12.67) |
| Merchant | 81 (36.3) | 142 (63.7) | 3.70 (0.81, 16.84) | 3.62 (0.67, 19.52) |
| Daily laborer | 6 (15.4) | 33 (84.6) | 1.18 (0.21, 6.62) | 2.25 (0.31, 16.17) |
| Others | 2 (13.3) | 13 (86.7) | 1 | |
| Exposure to mass media | | | | |
| Yes | 160 (51.8) | 149 (48.2) | 3.33 (2.18, 5.09) | 0.74 (0.38, 1.44) |
| No | 39 (24.4) | 121 (75.6) | 1 | |
| Perceived required time to reach health facilities | | | | |
| <30 minutes | 169 (46.3) | 196 (53.7) | 2.12 (1.32, 3.40) | 1.51 (0.79, 2.86) |
| ≥30 minutes | 30 (28.8) | 74 (71.2) | 1 | |
| Autonomy to health care decision making | | | | |
| Yes | 180 (47) | 203 (53) | 3.12 (1.80, 5.40) | 0.85 (0.40, 1.83) |
| No | 19 (22.1) | 67 (77.9) | 1 | |
| Knowledge of danger signs of pregnancy | | | | |
| Knowledgeable | 97 (51.9) | 90 (48.1) | 1.90 (1.30, 2.77) | 0.73 (0.44, 1.22) |
| Not knowledgeable | 102 (36.2) | 180 (63.8) | 1 | |
| Parity | | | | |
| 1–2 | 130 (49.2) | 134 (50.8) | 5.23 (2.55, 10.72) | 1.13 (0.34, 3.74) |
| 3–4 | 59 (41.8) | 82 (58.2) | 3.88 (1.83, 8.25) | 1.04 (0.32, 3.36) |
| ≥5 | 10 (15.6) | 54 (84.4) | 1 | |
| Pre-pregnancy utilization of contraceptive methods | | | | |
| Yes | 183 (48.9) | 191 (51.1) | 4.73 (2.66, 8.40) | 3.53 (1.80, 6.92) *** |
| No | 16 (16.8) | 79 (83.2) | 1 | |
| Planned pregnancy | | | | |
| Yes | 190 (51.1) | 182 (48.9) | 10.20 (4.99, 20.87) | 2.97 (1.27, 6.97) * |

*(Continued)*

**Table 4.** (Continued)

| Variable | Continuum of care | | COR (95% CI) | AOR (95% CI) |
|---|---|---|---|---|
| | Yes (%) | No (%) | | |
| No | 9 (9.3) | 88 (90.7) | 1 | |
| Timing of 1st ANC visit | | | | |
| Within 16 weeks | 132 (71) | 54 (29) | 6.42 (4.20, 9.80) | 4.57 (2.86, 7.31) *** |
| After 16 weeks | 67 (27.6) | 176 (72.4) | 1 | |

Others: farmer and driver, 1 = reference category

*significant at $0.01 \leq P \leq 0.007$

**significant at P = 0.005 and

***significant with P<0.0001

## Discussion

This study aimed to assess the coverage of the complete continuum of maternity care and associated factors. Accordingly, the coverage of a complete continuum of maternity care in the study area was 42.4% (95% CI: 37.9%, 47%).

The finding of the study was in agreement with a previous study done in Motta town and Hulet Eji Enese district 47% [23] and Enemay district 45% [24]. The possible reason for similarity of the study conducted in hulet ejui enese district and enemay district with the current study might be in study design similarity, all are done by community based cross sectional study designs and also the study populations' living conditions and cultural practices are remarkably comparable.

The results of this study were higher than those of studies conducted in Ethiopia West Gojjam Zone (12.1%) [13] and Arba Minch Zuria Woreda (9.7%) [12], in Ghana (8%) [17] and in Tanzania (10%) [6].The gap could be explained by differences in sampling technique, sample size, and data source. This difference might be due to; our study was conducted only in urban areas while the compared studies were done at a general level and specifically at rural areas which had relatively creates a better chance of accessing maternal health services. This was supported by studies showing that being rural resident negatively affect the chance of receiving full continuum of maternal care than urban. Additionally, the variation could be the result of different sampling methods, study periods and socio cultural differences.

The results of this study were lower than those of studies conducted in Egypt (50.4%) and Cambodia (60%) respectively [25, 28]. The differences might be explained by variation in geographic location, settings, access to health facilities, outcome ascertainment and eligibility criteria used. The inclusion of those who obtained at least one ANC or ANC4, and PNC within 48 hours or 6 weeks may have contributed to the variance compared to the current study. The other possible explanation could be that a longer study period retrospectively to assess the utilization that involved five years prior to the survey might increase their recall bias about the services they received for the last five years.

In this study, educated women had a better chance of receiving a complete continuum of maternity care than women with no formal education. Studies in Debre Berhan, South Asia and Sub-Saharan Africa, Pakistan, Cambodia, and Nepal backed with this conclusion [2, 15, 25, 27]. This could be explained by the fact that education can improve women's understanding, access to information, and capacity to grasp information about healthcare services from the media and healthcare workers. Furthermore, education may increase women's autonomy and assist them in developing better confidence and skill in making health-related decisions [28].

Women's who had recently used contraception had an increased likelihood of using the full range of maternal healthcare services. This result was consistent with studies done in Bangladesh [29] and Arba Minch Zuria Woreda [12], which discovered that pre-pregnancy contraceptive users utilized a continuum of maternal healthcare services more frequently [30]. This could be because women who used pre-pregnancy contraceptive planned ahead with a health care provider for future maternal and newborn services and were well-informed about those services.

Compared to their counterparts, women who planned for their recent pregnancy had a higher chance of receiving a complete spectrum of maternity care. This conclusion was supported by research undertaken at Arba Minch Zuria Woreda [12] and Debre Markos town [31]. A possible explanation is that having a planned pregnancy improves the likelihood of frequent visits, makes it less likely to delay commencement of care during pregnancy, and provides better information about the value of receiving maternity care throughout the pregnancy. Furthermore, women who had a planned pregnancy were far more cautious and eager to learn about their pregnancy status, and they were less likely to delay their initial ANC visit than those who had an unanticipated pregnancy.

In this study, higher odds of a complete continuum of maternity care were observed among participants who had early initiation of antenatal care during pregnancy compared to their counterparts. This finding was consistent with the result of studies conducted in West Gojjam Zone [13], Debre Berhan town [15], Arba Minch Zuria Woreda [12], and Kenya [32]. The reason for this could be that starting antenatal care early can help women get more antenatal visits and obtain important information about maternal and infant nutrition, birth, and emergency readiness. Another rationale is that pregnant women who attend ANC clinics early have a better chance of adapting to the health facility atmosphere, and health personnel avoid undue worry and stress by allowing women to freely express their health status with them.

## Strength and weakness of the study

The women who participated in the study were in the extended post-partum period since they were asked to recall their pregnancies and deliveries experience recall bias was possible. We tried to reduce recall bias by keeping the source population within 12 months of childbirth. The other potential bias in our study was social desirability bias Because the data were obtained by an interviewer-delivered questionnaire, which we attempted to address by using data collectors from various areas.

## Conclusion and recommendation

An integrated assessment of complete continuum of care from prenatal to postpartum stages was used in this study, and the results showed that the completion rate among women was only 42.4%, which is less than the recommended coverage of the world health organization and ministry of health. In this study, demographic and obstetric-related factors positively affected the complete continuum of maternity care. Factors like the educational status of women, pre-pregnancy contraception utilization, planned pregnancy, and timing of first antenatal care visit were statistically significant to a complete continuum of maternity care.

The results of this study draw attention to the need for interventions which increases maternal health promotion initiatives and health education, especially for low-educated women. Additionally, it is preferable to set up programs that encourage women to have early ANC visits, planned pregnancies, and improved access to pre-pregnancy contraceptive use, thereby increasing the coverage of complete continuum of maternal care.

## Supporting information

**S1 File.**
(DOCX)

## Acknowledgments

We would like to acknowledge Wollo University for ethical clearance and technical support as well as the Mekane Selam Health office and town administration for providing the necessary preliminary information. We would also extend our heart full gratitude to all research assistants and study participants for their genuine participation in this study.

## Author Contributions

**Conceptualization:** Hibist Tilahun Mengistie, Abrham Debeb Sendekie.

**Formal analysis:** Abrham Debeb Sendekie.

**Funding acquisition:** Hibist Tilahun Mengistie.

**Investigation:** Anguach Shitie.

**Methodology:** Hibist Tilahun Mengistie, Anguach Shitie.

**Project administration:** Hibist Tilahun Mengistie, Anguach Shitie.

**Resources:** Abrham Debeb Sendekie.

**Software:** Anguach Shitie.

**Supervision:** Mengistu Abate Belay, Dagne Addisu Sewyew.

**Validation:** Hibist Tilahun Mengistie, Mengistu Abate Belay.

**Visualization:** Mengistu Abate Belay, Abrham Debeb Sendekie, Dagne Addisu Sewyew.

**Writing – original draft:** Mengistu Abate Belay, Dagne Addisu Sewyew.

**Writing – review & editing:** Dagne Addisu Sewyew.

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
