## [Decision Letter · Decision Letter 0]

13 Jun 2023

PONE-D-22-15235Complete continuum of maternity care and associated factors among mothers who gave birth in the last twelve months in Mekane Selam town North-East Ethiopia: A Community-Based Cross-Sectional Study,2021.PLOS ONE

Dear Abraham Debeb Sendekie,

Thank you for submitting your manuscript to PLOS ONE. After careful consideration, we feel that it has merit but does not fully meet PLOS ONE’s publication criteria as it currently stands. Therefore, we invite you to submit a revised version of the manuscript that addresses the points raised during the review process.

Address the every reviewers comments in detail, clarify the method section, update the result and discussion section accordingly.

We look forward to receiving your revised manuscript.

Kind regards,

Tamirat Getachew

Academic Editor

PLOS ONE

3. In the ethics statement in the Methods, you have specified that verbal consent was obtained. Please provide additional details regarding how this consent was documented and witnessed, and state whether this was approved by the IRB

Additional Editor Comments:

Dear Authors, please revise your manuscript in detail and address each and every concerns of the reviewers. Mainly, you are expected to clarify the methods used for this study, the description of result section and discussion section.

Reviewers' comments:

Reviewer's Responses to Questions

**Comments to the Author**

1. Is the manuscript technically sound, and do the data support the conclusions?

Reviewer #1: No

Reviewer #2: Partly

Reviewer #3: Partly

Reviewer #4: Yes

2. Has the statistical analysis been performed appropriately and rigorously? 

Reviewer #1: Yes

Reviewer #2: Yes

Reviewer #3: Yes

Reviewer #4: Yes

3. Have the authors made all data underlying the findings in their manuscript fully available?

Reviewer #1: Yes

Reviewer #2: Yes

Reviewer #3: Yes

Reviewer #4: Yes

4. Is the manuscript presented in an intelligible fashion and written in standard English?

Reviewer #1: No

Reviewer #2: Yes

Reviewer #3: Yes

Reviewer #4: Yes

5. Review Comments to the Author

Reviewer #1: I doubt about the novelty of the title, since there were several published articles.i have also not seen any new variables which is different from other studies. All factors the authors identified were also known from previous studies. What new things you add in this study. Not yet identified.

Reviewer #2: Title: Complete continuum of maternity care and associated factors among mothers who gave birth in the last twelve months in Mekane Selam town North-East Ethiopia: A Community-Based Cross-Sectional Study,2021

The topic is relevant and is justified by the impact of childbirth care models on maternal mortality.

1. This manuscript is properly written and showed interesting findings.

2. The study presents the results of original research.

3. Conclusions are presented in an appropriate fashion and are supported by the data.

4. The article is presented in an intelligible fashion and is written in standard English.

5. The research meets all applicable standards for the ethics of experimentation and research integrity.

6. The authors provided valuable data and it is important to help governments, socialists and gynecologists in decreasing maternal mortality and fetal mortality and related diseases.

However, some questions and suggestions need to be made to further improvement of the manuscript. Authors reported that the list of the study participants was derived from the report of the zonal health office and health extension workers (HEWs). But it is very hard to get this list from health post (HEWs). They usually don't have the updated list, that cannot be true. The authors may miss women who deliver at their home. So, I have a big reservation here, unless you have more justification.

I suggest that discussion section start with a paragraph highlighting the most relevant accomplishments of the research. This section can be improved as follows.

1) Main findings in the first paragraph

2) Comparison with existing literature

3) Strengths and Weaknesses of the study should be clearly highlighted

4) Conclusion

Reviewer #3: PLOS ONE

Abstract

- You mentioned that: “Ethiopia remains one of the largest contributors to the global burden of maternal and neonatal deaths due to the low implementation of the complete continuum of maternity care”. If so, I think there is no need of conducting research rather evaluating the implementation and intervene accordingly.

- You use different words to describe complete continuum of maternity care (prevalence, magnitude….), since it is service utilization, it is better if you consider like proportion, coverage, utilization…

Introduction

- Contradicted ideas like: “zero maternal mortality” Versus “reducing maternal mortality to 70 maternal deaths per 100,000 live births”. Which one is correct?

- If the following studies are the same with your study, then why you are interested to repeat it? “According to a study based on EDHS 2016 data, only 9.1% of Ethiopian women completed the continuum of maternity care [11]. The magnitude of the complete continuum of maternity care ranges from 9.7% in Arba Minch Zuria Woreda [12], 12.1 percent in West Gojjam Zone [13], 21.6 percent in Gondar Zuria and Dabat districts [14], and 37.2

percent in Debre Berhan town, according to various studies conducted in Ethiopian rural districts [15]”. Even from the discussion part: “The study found that the prevalence of a complete continuum of maternity care was 42.4% (95% CI: 37.9%, 47%). The finding of the study was in line with the study done in Motta town and Hulet Eji Enese district 47% [23], Enemay district 45% [24], China 41.5% [25], and Nepal 45.7% [27]”.

- The last paragraph of the introduction (statement of the problem) is not convincing for me having the above studies.

Methods

- Sample size determination- why you use p=12.1% rather 37.2% which could give more than 590 study participants? So your sample is not adequate enough.

- Why multistage? Is that appropriate?

- How did you get the sampling frame for systematic sampling technique?

- “The questionnaire was written in English, then translated into Amharic (the local language) to collect data, and then back to English to ensure consistency”. Did you mean you translated back to English after data collection? How did this ensure the consistency?

Discussion

- Your reasoning is not scientifically sound and it is also poor.

Reviewer #4: Title: Complete continuum of maternity care and associated factors among mothers who gave birth in the last twelve months in Mekane Selam town North-East Ethiopia: A Community-Based Cross-Sectional Study, 2021

Comments:

1. Abstract section:

-Method – in method section you have to put the original sample and it was 479, and then in the result section you have to put the response rate

-The conclusion is not informative, therefore it need revision.

2. Introduction:

-Please revise paragraph three. The SDG goal is not exactly to make the maternal mortality 0, instead

“The SDG focus on the reduction of the global maternal mortality ratio to at least <70 per 100 000 live births or no country has no more than 140 maternal mortality ratio, reducing neonatal deaths to 12 per 1000 live births, and under-five deaths to <25 per 1000 live births through eliminating preventable maternal, neonatal and child deaths by the year 2030”

-Paragraph four: why you used EDHS 2016? Please use current data at least from EDHS 2019

-Again in paragraph four rewrite the following sentence - The magnitude of the complete continuum of maternity care ranges from 9.7% in Arba Minch Zuria Woreda [12], 12.1 percent in West Gojjam Zone [13], 21.6 percent in Gondar Zuria and Dabat districts [14], and 37.2 percent in Debre Berhan town, according to various studies conducted in Ethiopian rural districts [15] (please write your article as researcher)

-Please put % instead of the word percent after number (s)

-In the last paragraph, what do you mean the term ‘volume?’

3. Methods:

-in the study design and sitting you included the expected population of the town, please indicate for which year ___, and then put the reference

-your sampling procedure is not clear. How many kebeles existed in the town, and how did you select among them? Make it clear and try to show it in figure

-In sampling procedure I have a question – Which sampling technique is best for community based study? You have a sampling frame as you stated, so why did you used systematic sampling instead of simple random sampling technique?

-In the result section you stated the response rate was 97.9%, is it because of absent, if so what did you do? Or because of other issue. Please clarify it

-In operational definitions –You listed only four danger signs of pregnancy and you classified the women as knowledgeable on danger signs of pregnancy if she mentioned two or more danger signs you listed. Is swelling of legs is a danger signs of pregnancy? There are more danger signs of pregnancy, which you didn’t included in the list (it was best if you include all of the danger signs of pregnancy and assessed based on mean value)

-Data collection tools and quality measurements –please avoid redundancy (see it)

-Data processing –how many outcome variable did you have, make it ‘outcome variable’

-Ethical consideration – did you obtained both verbal and written consent? Choose one and include IRB number

4. Results:

-Avoid the term ‘respondent’ in your document – replace with the term ‘Women’

-And rewrite your results, b/c some of them didn’t give sense e.g. more than half of the 242 people who responded (51.6%) were between the ages of 25 and 34.

-In the table if the exact number is not 469, put in bracket after the variable. e.g., 337 of women have PNC visits, then you have to put like this ‘timing of PNC visits (n=337)

5. Discussion:

-See the 1st paragraph –replace the term study with studies

-Paragraph two - Discuss your result with more current information –use current EDHS data

-Paragraph 3 – did you include 6 weeks or 48 hrs to say completed continuum care of maternity? The explanation must be evidence based

6. Conclusion:

-The conclusion have to be based on your finding

-The recommendation should be based on the gab you identified, and achievable, and clear (please revise it)

Thank you!

6. PLOS authors have the option to publish the peer review history of their article (what does this mean?). If published, this will include your full peer review and any attached files.

Reviewer #1: No

Reviewer #2: **Yes: **Birhan Tsegaw Taye

Reviewer #3: No

Reviewer #4: **Yes: **Wondu Feyisa Balcha

---

## [Author Response · Author response to Decision Letter 0]

8 Jul 2023

We already upload a rebuttal letter that responds to each point raised by the academic editor and reviewers as a separate file labeled 'Response to Reviewers'.

---

## [Decision Letter · Decision Letter 1]

14 Jul 2023

Complete continuum of maternity care and associated factors among mothers who gave birth in the last twelve months in Mekane Selam town North-East Ethiopia: A Community-Based Cross-Sectional Study,2021.

PONE-D-22-15235R1

Dear Mr. Abraham Debeb Sendekie,

We’re pleased to inform you that your manuscript has been judged scientifically suitable for publication and will be formally accepted for publication once it meets all outstanding technical requirements.

Kind regards,

Tamirat Getachew

Academic Editor

PLOS ONE

Additional Editor Comments (optional):

Reviewers' comments:

Reviewer's Responses to Questions

**Comments to the Author**

1. If the authors have adequately addressed your comments raised in a previous round of review and you feel that this manuscript is now acceptable for publication, you may indicate that here to bypass the “Comments to the Author” section, enter your conflict of interest statement in the “Confidential to Editor” section, and submit your "Accept" recommendation.

Reviewer #2: All comments have been addressed

Reviewer #4: All comments have been addressed

2. Is the manuscript technically sound, and do the data support the conclusions?

Reviewer #2: Yes

Reviewer #4: Yes

3. Has the statistical analysis been performed appropriately and rigorously? 

Reviewer #2: Yes

Reviewer #4: Yes

4. Have the authors made all data underlying the findings in their manuscript fully available?

Reviewer #2: Yes

Reviewer #4: Yes

5. Is the manuscript presented in an intelligible fashion and written in standard English?

Reviewer #2: Yes

Reviewer #4: (No Response)

6. Review Comments to the Author

Reviewer #2: Thank you for adressing all my comments and suggestions. Hoppefully, this valuable data will help policy makers and

healthcare providers in decreasing maternal mortality and fetal mortality and related complications.

Reviewer #4: The authors have responded sincerely to the reviewers' comment and properly revised the manuscript.

7. PLOS authors have the option to publish the peer review history of their article (what does this mean?). If published, this will include your full peer review and any attached files.

Reviewer #2: **Yes: **Birhan Tsegaw Taye

Reviewer #4: **Yes: **Wondu Feyisa Balcha (wondufeyisaa85@gmail.com)

---

## [Editor Report · Acceptance letter]

18 Sep 2023

PONE-D-22-15235R1 

Complete continuum of maternity care and associated factors among mothers who gave birth in the last twelve months in Mekane Selam town North-East Ethiopia: A Community-Based Cross-Sectional Study,2021. 

Dear Dr. Sendekie:

I'm pleased to inform you that your manuscript has been deemed suitable for publication in PLOS ONE. Congratulations! Your manuscript is now with our production department. 

Kind regards, 

on behalf of

Dr. Tamirat Getachew 

Academic Editor

PLOS ONE